

Characterization of urban amine-containing particles in Southwestern China: seasonal
variation, source, and processing
Yang Chen[1,2*], Mi Tian[1], Rujin Huang[2], Guangming Shi[4], Huanbo Wang[1], Chao Peng[1],
Junji Cao[2], Qiyuan Wang[2], Shumin Zhang[3], Dongmei Guo[3], Leiming Zhang[5], and Fumo
Yang[1,4,*]
[1]Research Center for Atmospheric Environment, Chongqing Institute of Green and
Intelligent Technology, Chinese Academy of Sciences, Chongqing 400714, China.
[2]Key Lab of Aerosol Chemistry & Physics, State Key Laboratory of Loess and Quaternary
Geology, Institute of Earth Environment, Chinese Academy of Sciences, Xi'an 710061,
China.
[3]School of Basic Medical Sciences, North Sichuan Medical College, Nanchong 637000,
Sichuan, China.
[4]College of Architecture and Environment, Sichuan University, Chengdu 610065, China
[5]Air Quality Research Division, Science and Technology Branch, Environment and
Climate Change Canada, Toronto M3H 5T4, Canada
Correspondence to: Yang Chen (chenyang@cigit.ac.cn); Fumo Yang (fmyang@cigit.ac.cn)
Keyword: single particle; amine; urban environment, processing



**Abstract**
Amine-containing particles were characterized in an urban area of Chongqing during both
summer and winter using a single particle aerosol mass spectrometer (SPAMS). Among
the collected ambient particles, 12.7% were amine-containing in winter and 8.3% in
summer. Amines were observed to internally mix with elemental carbon (EC), organic
carbon (OC), sulfate, and nitrate. Diethylamine (DEA) was the most abundant in both
number and peak area among amine-containing particles. Wintertime amine-containing
particles were mainly from the northwest direction where a forest park was located; in
summer, they were from the northwest and southwest (traffic hub) directions. These origins
suggest that vegetation and traffic were the primary sources of particulate amines. The
average relative peak area of DEA depended strongly on humidity, indicating that the
enhancement of DEA was possibly due to increasing aerosol water content and aerosol
acidity. Using an adaptive resonance theory neural network (ART-2a) algorithm, four
major types of amine-containing particles were clustered including amine-organic-carbon
(A-OC), A-OCEC, DEA-OC, and A-OCEC-aged. The identified particle types imply that
amine was uptaken by particles produced from traffic and biomass burning. Knowledge
gained in this study is helpful to understand the atmospheric processing, origin, and sources
of amine-containing particles in the urban area of Chongqing.



## 1. Introduction

Amines are ubiquitous in the atmosphere and have both natural (ocean, biomass burning,
and vegetation) and anthropogenic (animal husbandry, industry, combustion, traffic)
emission sources (Ge et al., 2011a). Trimethylamine (TMA) is one of the most abundant
amines with an estimated global emission flux of 170Gg year$^{-1}$ (Ge et al., 2011a). Amines
in the gas phase compete with ammonia in acid-base reactions, participate in the gas-
particle partitioning, and contribute to wet and dry deposition (Angelino et al., 2001;
Monks, 2005; Gómez Alvarez et al., 2007; De Haan et al., 2011; Huang et al., 2012; You
et al., 2014). Gaseous amines also play an essential role in new particle formation via
enhancing the ternary nucleation of the sulfuric acid clusters in remote areas (Bzdek et al.,
2012; Kirkby et al., 2011). Recently, Yao et al. (2018) revealed that $H_2SO_4$-diethylamine
(DMA)-water clusters were important during the new particle formation events in polluted
urban areas. Amines are also essential in the growth of ambient particles. For example,
particulate aminium salts, which were produced via amine-acid neutralization, tended to
prevent the coagulation between pre-existing particles thus enhanced the particle number
concentration (Wang et al., 2010; Smith et al., 2010). Moreover, the enhancement of TMA
has been found during cloud and fog processing ((Zhang et al., 2012; Rehbein et al., 2011).
Understanding mixing state of amine-containing particles is important to understand their
processing and impact.
Single particle mass spectrometers (SPMS), such as aerosol time-of-flight mass
spectrometer (ATOFMS) and Single Particle Aerosol Mass Spectrometer (SPAMS), have
been used in real-time measuring amine-containing particles for chemical composition and



mixing state. The term SPAMS is different from the Aerodyne soot-particle aerosol mass
spectrometer (SP-AMS) which is a kind of aerosol mass spectrometer (AMS), detecting
the mass concentrations of black carbon, sulfate, nitrate, ammonium, chloride, and organics
(Onasch et al., 2012; Wang et al., 2016). The chemical composition and mixing state of
TMA-containing particles have been reported worldwide, such as in North America
(California, USA (Denkenberger et al., 2007; Qin et al., 2012)); Ontario, Canada (Tan et
al., 2002; Rehbein et al., 2011); Mexico City (Moffet et al., 2008)); Europe (Barcelona,
Cork, Zurich, Paris, Dunkirk and Corsica (Healy et al., 2015; Dall'Osto et al., 2016)), and
China (Guangzhou, Shanghai and Xi'an  (Zhang et al., 2012; Chen et al., 2016; Huang et
al., 2012)). Chemical composition and mixing state of amine-containing particles varied in
these locations. Thus the location-specific studies are still necessary.
Knowledge of amine-containing particles is limited in southwestern China. In this region,
Chongqing is a megacity with a population of 8.23 million and on the edge of the Sichuan
Basin. It is a subtropical, industrial, and polluted city (Chen et al., 2017b; Tao et al., 2017).
Fog events frequently occurred in this area, and the city is known as the "fog city" in China.
How high relative humidity (RH) affects the atmospheric processing of amine-containing
particles needs investigation. This study aims to characterize the amine-containing
particles, including chemical composition, mixing state, atmospheric processing, and
source in Chongqing during winter and summer.





**2. Methods**
**2.1 Sampling site**
Ambient single particles were collected at an urban air quality supersite from 07/05/2016
to 08/14/2016 (referred to as a summer season) and from 01/21/2016 to 02/25/2016
(referred to as a winter season). The supersite has been described in our previous studies
(Chen et al., 2017a; Chen et al., 2017b). Briefly, the supersite is located on the roof of a
commercial office building (106.51ºE, 29.62ºN) with a height of 30 m above the ground.
The building is surrounded by business and residential communities, 15 km away from the
city center. A forest park, with an area of 3 km$^2$, is located in the northwest of the sampling
site and a traffic hub in the southwest.
**2.2 Instrumentation**
A SPAMS was deployed for single particle sampling, and the technical description of the
instrument is available in literature (Li et al., 2011; Chen et al., 2017b). Briefly, after
passing through a diffusive dryer, particles in a size range of 0.1−2.0 μm are sampled via
an aerodynamic lens and form a particle beam. Particles in the beam cross two pre-
positioned laser beams (Nd: YAG, 532 nm) one-by-one, and the vacuum aerodynamic
diameter ($D_{va}$) of each particle is determined via its time-of-flight. Particles are ionized
using an Nd: YAG laser operating at a wavelength of 266 nm. The yielding ions are
analyzed using a bipolar time-of-flight mass spectrometer. Due to the limitation of SPAMS,
quantification of amines was not attempted.



### 2.3 Data analysis

The SPAMS data were imported into the YAADA toolkit (Software Toolkit to Analyze Single-Particle Mass Spectral Data, v 2.11) to form a particle dataset. The analysis was conducted using the marker ions of amines.: m/z 59 $[(CH_3)_3N]^+$ (TMA), 74 $[(C_2H_5)_2NH_2]^+$ (diethylamine, DEA), 86 $[(C_2H_5)_2NCH_2]^+$ or $[C_3H_7NHC_2H_4]^+$ (DEA or DPA), 101 $[(C_2H_5)_3N]^+$ (TEA), 102 $[(C_3H_7)_2NH_2]^+$ (DPA), 114 $[(C_3H_7)_2NCH_2]^+$ (DPA), and 143 $[(C_3H_7)_3N]^+$(TPA) (Healy et al., 2015). Firstly, m/z 59 was used for querying the TMA-containing particles; m/z 74 for the DEA-containing particles and m/z 86 for TEA-containing particles, and so on. After the duplicated particles in the query results being removed, these results were combined into an amine-containing particle cluster. Various amines could be both internally and externally mixed in these particle clusters.

An adaptive resonance theory based neural network algorithm (ART-2a) was applied to cluster the amine-containing particle types using a vigilance factor of 0.70, a learning rate of 0.05, and 20 iterations (Song et al., 1999). This procedure produced 67 clusters in summer and 75 clusters in winter; many of these clusters exhibited identical mass spectra with slight differences in specific ion peak areas. A well-established combining strategy, on the basis of similar mass spectra, temporal trends, and size distribution, was adopted to merge these particle clusters into the final particle types (Dallosto and Harrison, 2006).

Polar plot can gain an impression of the graphical distributions of potential sources influencing the measurement site. It presents concentration data of pollutants that vary by wind speed and wind direction (Carslaw et al., 2006).



## 3. Results and discussion

### 3.1 Single particle chemical composition and seasonal variation

Amine-containing particles were 12.7% in winter SPAMS dataset and 8.3% in the summer one. The DEA-containing particles were dominant among the total amine-containing particles, accounting for 70% and 78% in winter and summer, respectively, while TMA-containing particles were a minor group, accounting for up to 7% in winter and 3% in summer. The average mass spectra of DEA-, DPA, and TMA-containing particles are provided in Figure S1. All three mass spectra showed strong homogeneity. The determination coefficient ($R^2$) between DEA- and DPA- containing particles were 0.98, and $R^2$ between DEA- and TMA- containing particles was 0.83.

Figure 1 shows the digital mass spectra of amine-containing particles in two seasons. The assignment of ions is shown in Table 1. In both seasons, the dominant ions were $K^+$ (m/z 39 and 41), amines (m/z 59, 74, and 86), and organics (m/z 43, 51, 63, and 77). The mixing ratios of ammonium ($NH_4^+$, m/z 18) and polycyclic aromatic hydrocarbons (PAHs e.g., m/z 116 ($[C_9H_8]^+$), 129 ($[C_{10}H_9]^+$), 140 ($[C_{11}H_8]^+$), and 153 ($[C_{12}H_9]^+$)) were higher in winter than in summer. The strong signal of $NH_4^+$ was possibly due to the lower temperature (8°C) in winter than in summer (31°C). The mixing ratios of m/z 59 were 0.45 and 0.44 during summer and winter, respectively.

In the negative mass spectra of two seasons (Figures 1(b) and 1(d)), the dominating ions were $CN^-$ (m/z −26), $CNO^-$ (m/z −42), nitrate (m/z −46 and −62), phosphate (−79), and sulfate (m/z −80 and −97). Primary species, such as $CN^-$ and $CNO^-$ were commonly from





biomass burning (BB) and organonitrogen (Pratt et al., 2011). Levoglucosan markers from
BB, such as −45, −59, and − 71 were also detected. Dust markers, such as $[SiO_2]^-$ (m/z −
60), $[^{28}SiO_3]^-$ or $[AlO_2(OH)]^-$ (−76), and $[PO_3]^-$, were also detected during summertime,
suggesting the influence of dust particles.
Seasonal variations of chemical composition and unscaled size distribution are available in
supporting information. The average peak area of each ion was normalized in both summer
and winter, and the normalized values of each m/z in winter were used to subtract the
summer ones to generate a subtraction plot (Figure S2 (Qin et al., 2012)). A positive value
suggests that it is more predominant in summer, otherwise in winter. Nitrate was more
abundant in winter than summer. $Ca^+$ (m/z 40) and $Fe^+$ (m/z 56) were more prevalent during
summer. Organic species, such as $C_2H_3^+$ (m/z 27), $C_4H_3^+$ (m/z 51), $C_5H_3^+$ (m/z 63), and
$C_6H_5^+$ (m/z 77) typically from aromatic hydrocarbons, were more predominant in summer.
During wintertime, signals of sulfate (m/z −97), $NO_3^-$ (m/z −62), $NH_4^+$ (m/z 18), and $K^+$
(m/z 39) were more prominent than in summer, suggesting that the wintertime particles
contained more secondary species. The unscaled size distribution of amine-containing
particles also showed strong seasonal variations (Figure S3). Generally, amine-containing
particles had monomodal size distributions in the droplet mode; and the distributions
peaked at a larger $D_{va}$ in summer than winter. For example, DEA-containing particles
peaked at 0.6 µm in winter and 0.8 µm in summer, and DPA-containing particles at 0.7 µm
in winter and 0.9 µm in summer. The size distributions of the major amine-containing
particles suggested that these particles had undergone substantial aging processes.






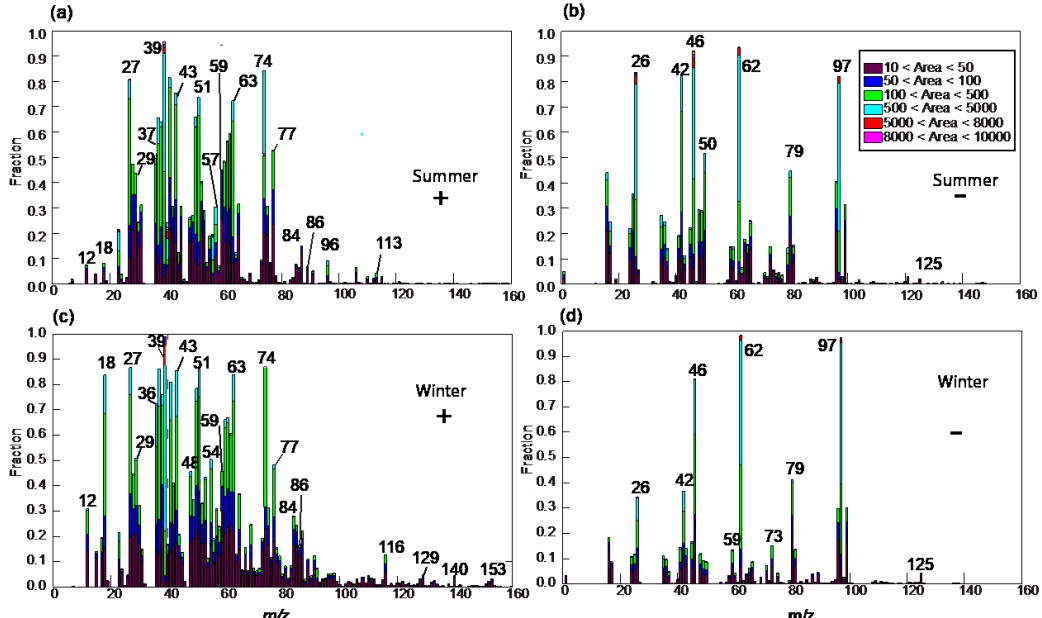

Figure 1. (a) and (c): the positive digital mass spectrum of amine-containing particles

during summer and wintertime, respectively; (b) and (d): the negative digital mass

spectrum during summer and wintertime, respectively. The ion height indicates its fraction

in the amine-containing particle dataset, and the stacked color map suggested the ion peak

area range.



Table 1. Assignment for ions in the mass spectra of amine-containing particles

| m/z | Ion assignment | m/z | Ion assignment |
|------|------|------|------|
| +12 | $C^+$ | −16 | $O^-$ |
| +18 | $[NH_4]^+$ | −17 | $[OH]^-$ |
| +23 | $Na^+$ | −26 | $[CN]^-$ |
| +24 | $Mg^+$ | −35 | $^{35}Cl^-$ |
| +27 | $Al^+/[C_2H_3O]^+$ | −37 | $^{37}Cl^-$ |
| +27 | $[CH_3N]^+/[C_2H_3]^+$ | −42 | $[CNO]^-$ |
| +30 | $NO^+$ | −43 | $[AlO]^-$ |
| +39 | $^{39}K^+$ | −46 | $[NO_2]^-$ |
| +40 | $Ca^+$ | −48 | $[SO]^-$ |
| +41 | $^{41}K^+$ | −50 | $[C_4H_2]^+$ |
| +43 | $[C_2H_3O]^+$ | −60 | $[AlO(OH)]^-$ or $[SiO_2]^-$ |
| +48 | $Ti^+$ | −62 | $[NO_3]^-$ |
| +51 | $C_4H_3^+$ | −63 | $[PO_2]^-$ |
| +54 | $^{54}Fe^+$ | −64 | $[SO_2]^-$ |
| +57 | $^{57}Fe^+$ or $[CaOH]^+$ | −73 | $[C_3H_5O_2]^-$ |
| +59 | $[(CH_3)_3N]^+$ | −76 | $[^{28}SiO_3]^-$ or $[AlO_2(OH)]^-$ |
| +63 | $[C_5H_3]^+$ | −77 | $[^{28}SiO_3]^-$ or $[H^{28}SiO_3]^-$ |
| +74 | $[(C_2H_5)_2NH_2]^+$ | −79 | $[PO_3]^-$ |
| +77 | $[C6H_3]^+$ | −80 | $[SO_3]^-$ |
| +113 | $[(CaO)_2H]^+$ | −88 | $[Si_2O_2]^-$ or $[FeO_2]^-$ |
| +116 | $[C_9H_8]^+$ | −97 | $[HSO_4]^-$ |
| +129 | $[C_{10}H_9]^+$ | −125 | $H[NO3]_2^-$ |
| +140 | $[C_{11}H_8]^+$ | | |
| +153 | $[C_{12}H_9]^+$ | | |





## 3.2 Temporal trend, diurnal pattern, and origin of amine-containing particles


Figure 2 shows the temporal tends of RH, temperature, number count, and peak area of
amine-containing particles. The winter temperature was lower (8.0±4.0 ℃) than summer
(31±4 ℃), and RH was slightly higher (70 ± 14% versus 64±16%) (Table 2). Stagnant air
conditions existed in both seasons due to the low wind speeds (Huang et al., 2017), and the
winter wind speed was lower than in summer. The hourly count of amine-containing
particles was mostly ten times higher in winter than summer.
Good correlations between the hourly number count and peak area were observed in the
temporal trends of DEA- ($R^2 = 0.86$) and DPA-containing particles ($R^2 = 0.88$) in winter.
No such correlation for TMA-containing particles was observed in winter ($R^2 = 0.22$) or
summer (Figure 2). Besides, the hourly counts between DEA- and DPA-containing
particles were well correlated in both summer ($R^2 = 0.63$) and winter ($R^2 = 0.87$), but only
weak correlation ($R^2 = 0.25$) existed between DEA- and TMA-containing particles. These
results suggest DEA- and DPA-containing particles were possibly from the same sources.
Table 2. Meteorological factors and particle counts in summer and winter.

| | Winter | Summer |
|---|---|---|
| Temperature (℃) | 8 ±4 | 31±4 |
| Relative humidity (%) | 70±14 | 64±16 |
| Wind Speed | 1.2±0.7 | 1.5±1.0 |
| Amin-particle Count (# h$^{-1}$) | 587±384 | 47±26 |




DEA- and DPA-containing particles remained at low levels from 1/20/2016 to 01/26/2016
and averaged at 109 and 26 count h$^{-1}$, respectively. During this period, wind speed was
relatively high, commonly above 1.5 ms$^{-1}$. TMA-, DEA-, and DPA-containing particles
started accumulating after 01/26/2016 when wind speed was low (0.8 ms$^{-1}$) and wind
direction from the northwest. After 02/03/2016, DEA- and DPA-containing particles
showed regular diurnal patterns with high levels of hourly count during the most daytime
and a minimum level at 15:00. A similar diurnal pattern was also observed for DPA-
containing particles during wintertime (Figures 3a and 3b). TMA-containing particles
presented a complex diurnal profile with peaks in the early morning (4:00), at noon (12:00)
and in the afternoon (18:00). The chemical composition and diurnal pattern of TMA-
containing particles were strongly connected to traffic emissions.
Wind direction and number count of amine-containing particles were analyzed together
using bivariant polar plots (Figure 4). During wintertime, the dominant origin for amine-
containing particles was from the northwest where a forest park was located. After being
emitted from vegetation  (plants, grass, and trees) (Norton, 1985), DEA partitioned to the
pre-existing particles. These particles were transported to the sampling site, causing the
elevation in the morning. Based on the excellent correlation between DEA- and DPA-
containing particles, DPA-containing particles could also be from vegetation. It can be
concluded that vegetation was the major source of amines in DEA- and DPA-containing
particles from the northwest.

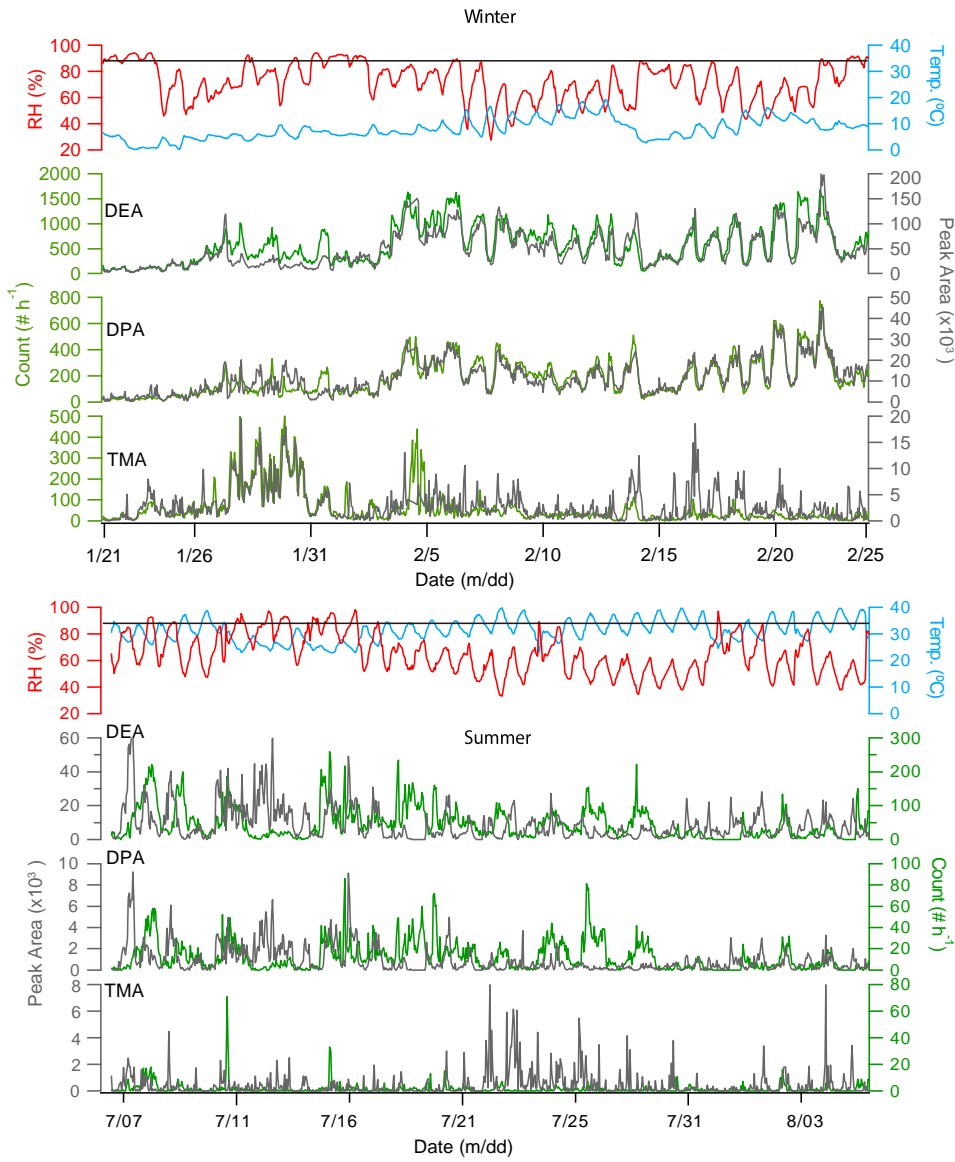


Figure 2. Temporal trends of relative humidity (RH), temperature (Temp.), hourly peak
area (dark gray), and particle count (green) of DEA (m/z 74), DPA (m/z 86), and TMA
(m/z 59) -containing particles in winter (top panel) and summer (bottom panel). The black
lines in two panels indicate RH of 90%.



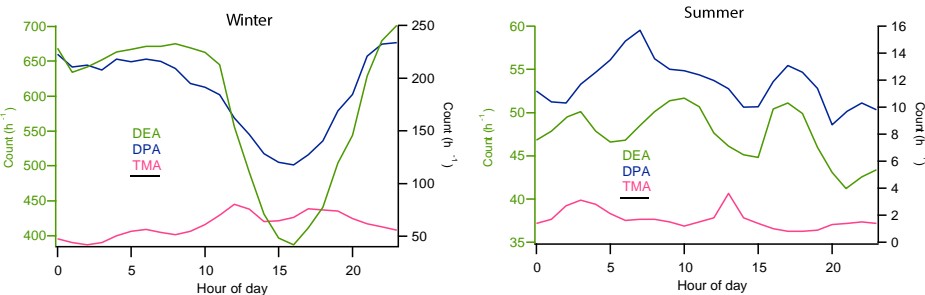


Figure 3. Diurnal profiles of amine-containing particles during both winter (left panel) and summer (right panel). The green left axis in each panel indicates the average number count of DEA-containing particles, while the right-axis represents the number count of both DPA- and TMA-containing particles.

During summer, the amine particles appeared in several episodes; each episode lasted for 1~3 days. In these episodes, DPA-containing particles had two rush-hour peaks (7:00 and 17:00), likely because they were also produced from traffic (Dall'Osto et al., 2016); besides the vegetation is a source of DPA-containing particles (from the southwest, Figure 4e). The DPA-containing particles peaked 0.84 µm, suggesting that they were not freshly-emitted and had undergone substantial aging processes.

In summer, DEA-containing particles had a diurnal pattern of three peaks appearing at 3:00, 9:00 and 17:00. TMA-containing particles had an early morning (4:00) and a noon peak (12:00). The morning peaks of DEA- and TMA-containing particles could be due to the local traffic activities, specifically, the heavy-duty vehicles which were only allowed to enter the urban area between 00:00-6:00 (Chen et al., 2017b). The polar plot showed that DEA-containing particles were from the northwest and southwest, passing through the forest park and traffic hub, respectively. This scenario seemed to be inconsistent with the



wintertime results because the traffic contributed limitedly in winter. In addition, due to
the competition between vegetation and traffic, number count, and peak area of all the three
amine-containing particles were poorly correlated with each other in summer.

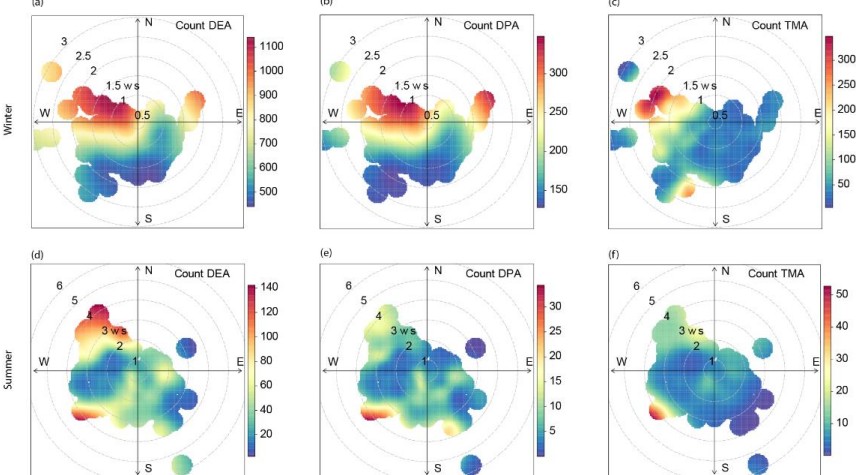


Figure 4. Polar plots of the count of amine-containing particles during winter- and
summertime. The circles in each figure indicate wind speed (ws).
**3.3 Effect of RH on the enrichment of DEA-containing particles**
DEA-containing particles were predominant in both winter and summer, providing a
unique opportunity for investigating DEA processing. Indeed, this kind of discussion
should be treated cautiously and the influences of wind speed, wind direction, temperature,
and planetary boundary layer reduction should be removed. As described above, average
wind speed in both winter and summer was 1.2 ms$^{-1}$ and 1.5 ms$^{-1}$, respectively. In these
stagnant air conditions, the sampled particles were generally local. Temperature could
influence the gas-particle phase portioning. Assuming the Henry's Law constants ($K_H$) and



the enthalpy change $\Delta_r H_o (K_H)$ of DEA are constant, a variation of 10ºC in both summer
and winter could negligibly influence the portioning of amines from the gas phase to the
particle phase, according to the Clapeyron equation (Ge et al., 2011b). In addition, the shift
of planetary boundary layer (PBL) height could affect the number count and concentration
of PM. The relative peak area (RPA) is defined as the peak area of each m/z divided by the
total dual-ion mass spectral peak areas in an settled time bin (typically one hour) (Healy et
al., 2013). Using the temporal trends of RPA can remove the influence of PBL height
because it only shows the relative changes between different species which are
simultaneously influenced by the shift of PBL height.
Box plots of DEA relative peak area under different RH conditions are shown in Figure 5.
In winter, the median RPA of amine-containing particles increased by two times when RH
increased from 35% to 95%. Meanwhile, the fraction of DEA-containing particles
increased from 4.0% to 16.6%. In summer, the average RPA of DEA increased by three
times (from 0.25 to 0.75) and the fraction of DEA-containing particles ramped from 3.8%
to 12.1% when RH increased from 60% to 90%. These results suggest that RH is important
for the enrichment of DEA in the particle phase. DEA was favorable to form DEA salts
when reacting with HCl, $H_2SO_4$, and $HNO_3$, and these salts had good solubility in water,
making them easy to enter the aerosol phase. Along with the influence of aerosol water
content, Ge et al. (2011a) also proposed that strong aerosol acidity could also enhance the
partitioning of DEA in the aqueous phase. In this study, the relative acidity of amine-
containing particles ((sulfate +nitrate)/ammonium, (Yao et al., 2011)) was in a range of 20-
150, providing favorable conditions for the dissolution of DEA. Indeed, due to the nature
of SPAMS, the amount of aerosol water content and pH were unavailable, making it



difficult for further analysis. Overall, these results imply that high RH condition in
Chongqing was favorable for DEA to go uptake on particles, and the formed aminium salts
which stabilized pre-existing particles and increased their number concentrations.
Rehbein et al. (2011) and Zhang et al. (2012) observed direct links between fog processing
and the enhance of TMA-containing particles. High RH conditions were favorable for
TMA entering the particle phase via gas-to-particle partitioning (Rehbein et al., 2011;
Zhang et al., 2012). Ge et al. (2011b) argue that TMA in the aerosol phase was in the form
of free base, e.g., amine, not aminium salt; TMA could be dissolved in the aerosol water
content; the formation of $TMA\text{-}HSO_4$ salt was possible, but the formation of $TMA\text{-}NO_3$
and $TMA\text{-}Cl$ was impossible due to the competition of ammonia. Thus, TMA could enter
the aerosol phase by gas-aqueous partitioning, or in the form of $TMA\text{-}HSO_4$ salt. The
mechanism of DEA entering the aerosol phase might be different from TMA. DEA salts
were easy to form (Ge et al., 2011b). Besides, Pankow (2015) proposed that the absorptive
uptake of atmospheric amines could also be possible on organic aerosols. In the context of
single particle mixing state, the amine-containing particles were internally mixed with
hygroscopic species, e.g., sulfate and nitrate, POA species ($C_xH_y^+$, see section 3.4), and
SOA species (oxalate, $C_2H_3O^+$). Therefore, the mixing state of amine-containing particles
was also favorable for the uptake of amines via different pathways: the aqueous dissolution
of aminium salts, absorptive uptake on POA and SOA.





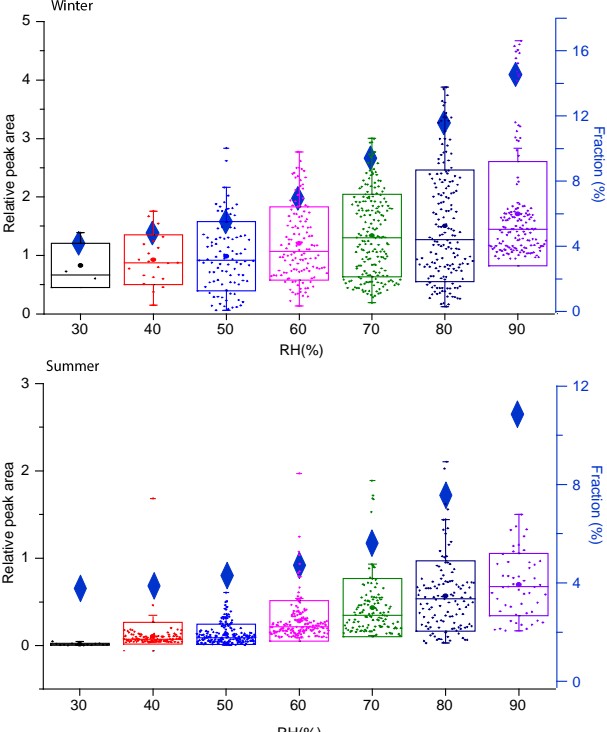


Figure 5. Box plots of the hourly relative peak area of DEA under different RH conditions
in winter (top panel) and summer (bottom panel). The boxes indicate the 25th and 75th
percentiles; the dots indicate mean value with each data point representing a datum of RPA
in an hour size bin. Right axis in each panel and the blue diamond show the average number
fraction of amine-containing particles among the whole SPMAS dataset.
**3.4 Particle types of amine-containing particles**
As shown in Figure 6, four types of amine-containing particle types were resolved,
including amine-OC (A-OC, 41%), A-ECOC (39%), DEA-OC (11%), and A-ECOC-aged
(9%). All these particle types had strong signals of amines, and the amines were internally
mixed with sulfate, nitrate, elemental carbon, and organics.





In the A-OC particles, the amines were present with aromatic hydrocarbon fragments, such
as $C_4H_3^+$ (m/z 51), $C_5H_3^+$ (m/z 63), $C_6H_5^+$ (m/z 77), and $C_9H_8^+$ (m/z 116), as well as with
alkanes fragments such as $C_4H_7^+$ (m/z 55), $C_4H_9^+$ (m/z 57), and $C_5H_9^+$ (m/z 69). In the
negative mass spectrum of A-OC, strong signals from $CN^-$ (m/z −26 ) and $CNO^-$ (m/z −42)
were typically primary species, along with levoglucosan (Silva et al., 1999). The amine
fragments, such as TMA (m/z 59), DEA (m/z 74), and DPA (m/z 86) were well abundant
in this particle type (76%, 95%, and 88%, respectively). The parent particles of A-OC was
a kind of OC particles from biomass burning; then they mixed with amines via uptake.
Meanwhile, A-OC could also be hydrophilic due to the presence of sulfate and nitrate,
which could turn the particle into the water phase, making it possible for the dissolution of
amines.
In A-ECOC mass spectra, strong signals of amines (m/z 59 and 74), along with the major
aromatic hydrocarbon fragments mentioned above were detected. In the negative mass
spectra, nitrate and sulfate were also dominant. The A-ECOC-aged particle type had a
similar chemical composition to A-ECOC ($R^2$ =0.53) but with the weaker relative
intensities of $C_xH_y^+$ and amine ions, suggesting it could be more secondary.
In the positive mass spectra of DEA-OC, DEA fragment (m/z 74) was dominating and
presenting with organic fragments described above. The secondary organic marker ions,
such as m/z 43 ($[C_2H_3O]^+$) and −89 (oxalic acid) were found in the mass spectra. Besides,
DEA-OC is not sensitive to wind speed ($R^2$ =0.18), implying they were local.




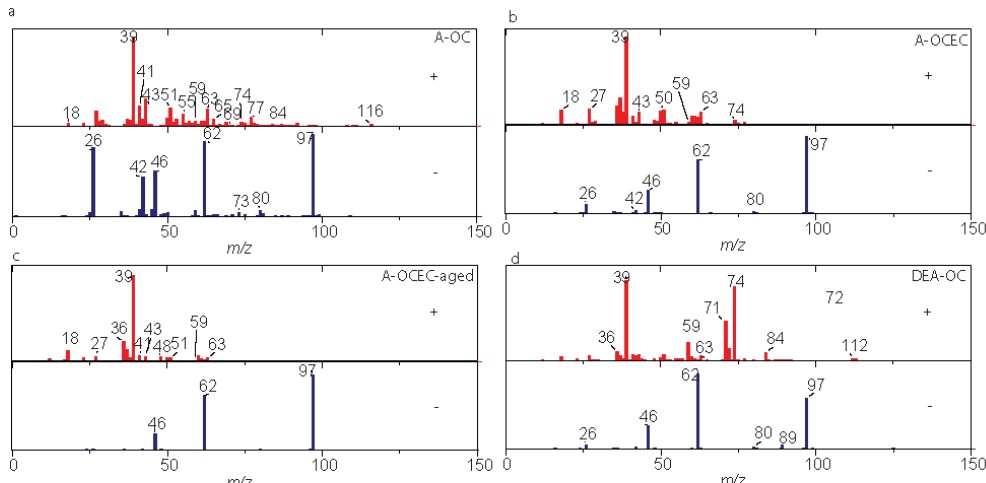


Figure 6. Average mass spectra of major particle types clustered from amine-containing

particles.

The summertime amine-containing particles were similar to the particle types during winter

(all $R^2$ >0.7) except that a Ca-rich particle type was also resolved (Figure S4). A-Ca-OC

particle type was majorly composed of Calcium ($Ca^+$ and $CaO^+$), potassium (m/z 23), TMA

(m/z 59), sulfate, nitrate, and phosphate. The A-Ca-OC particle type was from traffic

activities (Chen et al., 2017a).

The amine-containing particle types reported in this study were different from those in

literature. Cheng et al. (2018) reported that m/z 74 amine-containing particles were most

abundant in the Pearl River Delta, China, but the chemical composition and mixing state

of amine particles were different from the present study. For example, the mixing ratio of

DPA was much stronger (~0.2) than in this study (<0.1). In most related studies, TMA-

containing particles were dominant while the present study showed DEA-containing





particles were dominant (Rehbein et al., 2011; Zhang et al., 2012; Healy et al., 2015; Dall
et al., 2016).
**4. Conclusions**
The amine-containing particles were analyzed using a SPAMS during winter and summer
in the urban area of Chongqing. Generally, amine-containing particles were more abundant
in winter than in summer. DEA-containing particles (m/z 74) were the most important
particle type during two observation periods. The amine-containing particles were mostly
from vegetation located southwest of the sampling area. An enrichment of DEA-containing
particles under high RH conditions was revealed. Amines were commonly mixed with
elemental carbon, organics, sulfate, and nitrate. The amine-containing particles
substantially aged during the transport. Reduction of anthropogenic amines such as DEA
and TMA would improve the air quality in this region, which can be achieved by decreasing
emissions of the on-road fuel-powered automobiles.
Acknowledgments. Financial support from the Nature Science Foundation of China (Grant
No. 41375123), the National Key Research and Development Program of China
(2016YFC0201506 and 2018YFC0200403), and the Starting-up project for Ph.D.
(15ZA0213) are acknowledged.
Author Contribution. CY and YF designed the experiments; TM, SG, PC, WH, and WQ
carried them out; HR, CY, ZL, CJ, and GD analyzed the experiment data; CY prepared the
manuscript with contributions from all co-authors.



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
