# Peer review of "Characterization of urban amine-containing particles in Southwestern China: seasonal"

_Atmospheric Chemistry and Physics, 2018_

## Referee Comment (RC1) · Anonymous Referee #2 · 31 Dec 2018

Chen et al. investigated amine-containing particles in southwestern China, including the chemical composition, mixing state, source, and processing. The authors described the processing of DEA-containing particles, suggesting that high relative humidity conditions were favorable for the enrichment of DEA in the particle phase. This finding is of novelty, and the characterization of amine-containing particles in southwestern China is useful for improving the scientific understanding of the atmospheric processing of amine-containing particles in this area. The manuscript is Weill structured and drafted, but still with some grammar and syntax issues. The referee would recommend publishing when some minor issues are addressed. Major issue Section 3.4, please compare your results with the studies worldwide and describe the

differences in various locations. Conclusion: the effect of relative humidity on DEA-containing particles should be mentioned. Specific issues, Line 68-69, how these amine-containing particles varied? Please briefly describe them. Section 2.1 A map of sampling site would be helpful. Line 108-109, the sentence is unclear, please reword it. Line 118, "impression" or "expression?" Line 172, the effect of air stagnant should be addressed, which will be helpful to understand the atmospheric processing of amine-containing particles. Line 220-225, according to Figure 4, what the possible source for DPA-containing particle from the northeast? Line 264 and 282, "easy" is an informal term, please change it into a formal one. Line 267, please put the comma after (Yao et al. 2011) Line 306, "was" should be "were."

---

## Referee Comment (RC2) · Anonymous Referee #1 · 8 Jan 2019

Overall, this paper describes the observed amine-containing atmospheric aerosol particles sampled with a single particle mass spectrometer in Chongqing. The authors make the case that amine-containing particles are different in various locations in which they have been sampled globally, and therefore that it is necessary to investigate the amine-containing particles in every possible location for potential new insights. This paper describes the amine-containing particle types observed, the dependence of the amine components on air-mass origin, the dependence on relative humidity, etc. Overall, the paper provides a good snapshot of amine chemistry in this location in two seasons. It needs to be thoroughly proofread and the grammar corrected throughout – the edits are mostly relatively minor and will not be enumerated here. Specific Comments:

[Figure]

Line 75: The authors state that "how high relative humidity (RH) affects the atmospheric processing of amine-containing particles needs investigation." They should elaborate about why this investigation is necessary – what does one learn from it? Section 2.1: A map would be helpful, especially in interpretation of the polar plots shown later. It could be in the SI. Line 90: The instrument needs to be described. If it is a commercial instrument, provide vendor and model. If it is laboratory-built, indicate this. I presume it is the former. Line 103 and subsequent uses: When m/z is written, both the m and the z should be italicized, but the / should not be, to conform with mass spectrometry standards. Lines 118 – 120: This is a standard data plot format and does not need to be described in the methods section. Line 123: Is the percentage quoted here (12.7% and 8.3%) the percent of all particles observed in the respective season? This should be clarified. Line 131: "Digital mass spectrum" needs to be defined in the text, not just the caption of Figure 1. Line 137: When referring to "mixing ratio," are the authors suggesting that 44% and 45% of particles in summer and winter, respectively, contained a peak at m/z 59? If so, just say that. Lines 147 – 150 and Figure S2: The description in the text doesn't match the figure. It should say that the normalized ion intensity of the winter-time particles was subtracted from that of the summer-time particles, and that a positive value indicates the normalized ion intensity was greater in the summer, whereas a negative value indicates that the normalized ion intensity was greater in the winter. Also, the authors should specify how the peak area was normalized. Line 162: Was this aging occurring in both seasons? Line 167: Replace "suggested" with "indicates." Table 1: This should go into the SI, as these ion assignments are common in the single-particle mass spectrometry literature. Line 176: Should "mostly" be "typically"? I am not certain the meaning of this sentence, otherwise. Line 188: "Amine" should have an e on the end. Figure 2: If there are specific times that the authors want to draw the readers' attention to, highlighting the range of days would be helpful. Lines 220 – 225: This section is somewhat confusing. The authors seem to be saying that there are two sources for DPA-containing particles, but more evidence should be cited from the mass spectra, not just the direction from which the wind was blowing.

Line 252: Isn't RPA usually defined on a per-particle basis, rather than within a time bin? How this is calculated should be clearly stated, earlier in the paper. Lines 262 – 264: The statement about DEA is confusing and should be reworded. Line 269: If the particles were sampled through a drier, as stated in the methods, then of course no information about particle water content will be available. Line 294: It should read "SPAMS" rather than "SPMAS." Lines 296 – 299: This section would make more sense before the discussion in lines 273 – 288. Line 309: Reword this, as it doesn't make a lot of sense. Line 311 and below, and Figure 6: The authors include A-ECOC as a particle type, but it is not obvious where the EC components are – the typical EC ions are not visible. This should be clarified. Line 325: The authors mixed up Na+ and K+ when they refer to "potassium (m/z 23)." It seems that either ion could be referred to here. Line 326: The Ca-containing particle that is described looks a lot like dust. Can the authors make a strong case that it is traffic and not dust? Line 341 and 346: The authors need to say more about how they are making the case that certain amines are both from vegetation and traffic. In the text, they refer to DPA being from both sources, and here in the conclusions they refer to DEA. Section 4: The conclusions do not summarize the conclusions made throughout the paper and should be expanded.
* * *

---

## Author Comment (AC1) · 26 Feb 2019

Dear referee,

The authors are thankful for the reviewer's efforts on the valuable comments for this manuscript. These comments would be very helpful for us to improve the manuscript. We would like to provide a point-to-point response. A native English-speaking Scientist did the proofreading for the text.

Major issue Section 3.4, please compare your results with the studies worldwide and describe the differences in various locations. Conclusion: the effect of relative humidity

on DEA-containing particles should be mentioned. We sincerely appreciate this comment. Yes, this part has been expanded into a broader context, please see the revised Section 3.4.

Specific issues,

Line 68-69, how these amine-containing particles varied? Please briefly describe them.

Yes, we have expanded this part in the revised version (line 72-79): "In the five European cities such as Cork, Paris, Dunkirk, Corsica, and Zurich, amines were found internally mixed with sulfate and nitrate; but in Corsica, amines were internally mixed with methanesulfonate (Healy et al., 2015). In Barcelona, five unique types of amine-containing particles were observed (Dall'Osto et al., 2016). In a rural area site in the Pearl River Delta (China), the marker ion, $(C2H5)2NH2+$, was the most abundant (90% and 86% of amine-containing particles in summer and winter)(Cheng et al., 2018). In Guangzhou, TMA-containing particles were important (Zhang et al., 2012). "

Section 2.1 A map of sampling site would be helpful.

A map has been added in the supportive information (Figure S1).

Line 108-109, the sentence is unclear, please reword it.

The sentence has been modified (line 121-123):

"After the duplicate particles were removed from the query results, all amine-containing particles were combined into an amine-containing particle cluster."

Line 118, "impression" or "expression?"

It should have been "expression," we have changed it.

Line 172, the effect of air stagnant should be addressed, which will be helpful to understand the atmospheric processing of amine-containing particles.

We have modified the text into: "Such low wind speed caused stagnant air conditions

in both summer and winter. "

Line 220-225, according to Figure 4, what the possible source for DPA-containing particle from the northeast?

It was also from traffic; please see the revised discussion (line 251).

Line 264 and 282, "easy" is an informal term, please change it into a formal one.

The comment has been accepted and the change made (Line 313).

Line 267, please put the comma after (Yao et al. 2011)

The comment has been accepted and the change made.

Line 306, "was" should be "were."

The comment has been accepted and the change made (line 340).

Reference

Cheng, C. L., Huang, Z. Z., Chan, C. K., Chu, Y. X., Li, M., Zhang, T., Ou, Y. B., Chen, D. H., Cheng, P., Li, L., Gao, W., Huang, Z. X., Huang, B., Fu, Z., and Zhou, Z.: Characteristics and mixing state of amine-containing particles at a rural site in the Pearl River Delta, China, Atmos Chem Phys, 18, 9147-9159, 10.5194/acp-18-9147-2018, 2018.

Dall'Osto, M., Beddows, D. C. S., McGillicuddy, E. J., Esser-Gietl, J. K., Harrison, R. M., and Wenger, J. C.: On the simultaneous deployment of two single-particle mass spectrometers at an urban background and a roadside site during SAPUSS, Atmos Chem Phys, 16, 9693-9710, 10.5194/acp-16-9693-2016, 2016.

Healy, R. M., Evans, G. J., Murphy, M., Sierau, B., Arndt, J., McGillicuddy, E., O'Connor, I. P., Sodeau, J. R., and Wenger, J. C.: Single-particle speciation of alkylamines in ambient aerosol at five European sites, Anal Bioanal Chem, 407, 5899-5909, 10.1007/s00216-014-8092-1, 2015.

Zhang, G., Bi, X., Chan, L. Y., Li, L., Wang, X., Feng, J., Sheng, G., Fu, J., Li, M., and Zhou, Z.: Enhanced trimethylamine-containing particles during fog events detected by single particle aerosol mass spectrometry in urban Guangzhou, China, Atmos Environ, 55, 121-126, 10.1016/j.atmosenv.2012.03.038, 2012.

Please also note the supplement to this comment:
https://www.atmos-chem-phys-discuss.net/acp-2018-1119/acp-2018-1119-AC1-supplement.pdf
* * *

---

## Author Comment (AC2) · 26 Feb 2019

Dear Reviewer,

The authors are sincerely thankful for your valuable comments and suggestion regarding this manuscript, and we appreciate the positive comments and encouragement from the reviewer. We prefer to respond to your comments and questions point-to-point. The manuscript has been proofed by a native English-speaking Scientist from Canada.

Interactive Comment on "Characterization of urban amine-containing particles in

[Figure]

Southwestern China: seasonal variation, source, and processing" by Yang Chen et al. Anonymous Overall, this paper describes the observed amine-containing atmospheric aerosol particles sampled with a single particle mass spectrometer in Chongqing. The authors make the case that amine-containing particles are different in various locations in which they have been sampled globally, and therefore that it is necessary to investigate the amine-containing particles in every possible location for potential new insights. This paper describes the amine-containing particle types observed, the dependence of the amine components on air-mass origin, the dependence on relative humidity, etc. Overall, the paper provides a good snapshot of amine chemistry in this location in two seasons. It needs to be thoroughly proofread and the grammar corrected throughout – the edits are mostly relatively minor and will not be enumerated here.

Specific Comments:

Line 75: The authors state that "how high relative humidity (RH) affects the atmospheric processing of amine-containing particles needs investigation." They should elaborate about why this investigation is necessary – what does one learn from it?

The incoherence has been fixed with a statement in the last paragraph (Line 79-82): In previous studies, reported high RH conditions and fog processing were favorable for the enhancement of trimethylamine in the particle phase. Zhang et al. (2012) reported a similar scenario in Guangzhou, China.

Section 2.1: A map would be helpful, especially in interpretation of the polar plots shown later. It could be in the SI.

Yes, we have appended a sampling site map in the SI and cited in the text (Line 99).

Line 90: The instrument needs to be described. If it is a commercial instrument, provide vendor and model. If it is laboratory-built, indicate this. I presume it is the former.

We have provided the information on the manufacturer and model number (Hexin Inc. Guangzhou, China, model: 0515) in Section 2. 2 (Line 103).

Line 103 and subsequent uses: When m/z is written, both the m and the z should be italicized, but the / should not be, to conform with mass spectrometry standards.

We have changed the style of m/z in the text.

Lines 118 – 120: This is a standard data plot format and does not need to be described in the methods section.

Accepted and changes have been made.

Line 123: Is the percentage quoted here (12.7

We have changed the sentence for clarity (Line 138-140):

"The percentage of amine-containing particles was 12.7

Line 131: "Digital mass spectrum" needs to be defined in the text, not just the caption of Figure 1.

The definition has been added into the text.

Line 137: When referring to "mixing ratio," are the authors suggesting that 44

Affirmative, we have changed this part.

Lines 147 – 150 and Figure S2: The description in the text doesn't match the figure. It should say that the normalized ion intensity of the winter-time particles was subtracted from that of the summer-time particles, and that a positive value indicates the normalized ion intensity was greater in the summer, whereas a negative value indicates that the normalized ion intensity was greater in the winter. Also, the authors should specify how the peak area was normalized.

The authors are very thankful for this instruction. And we have changed the sentence (Line 164-170): " Prior to comparison, the ion peak was normalized using the method developed by Qin et al. (2012). Briefly, the peak area of each m/z was divided by the total mass spectral peak area matrix. The normalized ion intensity of the wintertime

particles was subtracted from that of the summertime particles. A positive value indicates the normalized ion intensity was greater in the summer, whereas a negative value indicates that the normalized ion intensity was greater in the winter."

Line 162: Was this aging occurring in both seasons?

Yes, the amine-containing particles were aged in both two seasons. In previous studies, a peaking larger than 0.5 $\mu$m suggested that the particle type was aged (Chen et al., 2017).

Line 167: Replace "suggested" with "indicates."

It is accepted and changed (line 187).

Table 1: This should go into the SI, as these ion assignments are common in the single-particle mass spectrometry literature.

It is accepted and changed.

Line 176: Should "mostly" be "typically"? I am not certain the meaning of this sentence, otherwise.

It should have been "typically". Amine-containing particles were more abundant during winter than summer (line 195).

Line 188: "Amine" should have an e on the end.

We have changed this typo.

Figure 2: If there are specific times that the authors want to draw the readers' attention to, highlighting the range of days would be helpful.

Thank you for this suggestion, we do not have a specific period to address, but tried to introduce a general description in both two seasons.

Lines 220 – 225: This section is somewhat confusing. The authors seem to be saying that there are two sources for DPA-containing particles, but more evidence should be

cited from the mass spectra, not just the direction from which the wind was blowing.

We have added more description and citation into this paragraph (Line 247-253).

"Moreover, as shown in Figure S2, the mass spectra of the amines were present with aromatic hydrocarbon fragments, such as C4H3+ (m/z 51), C5H3+ (m/z 63), C6H5+ (m/z 77), and C9H8+ (m/z 116), as well as with alkanes fragments such as C4H7+ (m/z 55), C4H9+ (m/z 57), and C5H9+ (m/z 69). The chemical composition of DPA-containing particles contained markers associated with traffic emissions. In addition, a similar amine-containing particle type has been reported in the literature (Dall'Osto et al., 2016)."

Line 252: Isn't RPA usually defined on a per-particle basis, rather than within a time bin? How this is calculated should be clearly stated, earlier in the paper.

We have fixed this issue and added this part of instruction into Section 2.3. Please see line 134-135 for details.

Lines 262 – 264: The statement about DEA is confusing and should be reworded.

We have modified the sentence into: When DEA reacts with HCl, H2SO4, and HNO3, it tends to form aminium salts, which are soluble in aerosol water (Line 293-294).

Line 269: If the particles were sampled through a drier, as stated in the methods, then of course no information about particle water content will be available.

We are sorry to say that there was no information on aerosol water content, and we have specified this in the text (Line 299-301):

"Indeed, due to the nature of SPAMS, the amount of aerosol water content and pH were unavailable, making it difficult for further analysis."

Line 294: It should read "SPAMS" rather than "SPMAS."

We have changed this typo (Line 327).

Lines 296 – 299: This section would make more sense before the discussion in lines 273 – 288. Since we combined the Results and discussion, this adjustment could probably influence the flow of the text. We prefer an independent section for this part.

Line 309: Reword this, as it doesn't make a lot of sense.

We have modified this part of the sentence (Line 339-341)

"Amines could enter the A-OC particle type via dissolution in the aerosol water content or uptake due to absorptive uptake on the OC aerosol (Pankow, 2015)."

Line 311 and below, and Figure 6: The authors include A-ECOC as a particle type, but it is not obvious where the EC components are – the typical EC ions are not visible. This should be clarified.

We have modified Figure 6 and sentence, adding the label of EC component (i.e., m/z 36, 48, 60) for clarity (Line 345).

Line 325: The authors mixed up Na+ and K+ when they refer to "potassium (m/z 23)." It seems that either ion could be referred to here.

We have modified this sentence, adding both ions into the text (Line 359-360).

Line 326: The Ca-containing particle that is described looks a lot like dust. Can the authors make a strong case that it is traffic and not dust?

Yes. An ion signal of zinc (m/z 64) was observed in the positive mass spectrum. Zn is a marker for tire wear on road (Grigoratos and Martini, 2015; Thorpe and Harrison, 2008). Thus, the authors preferred the particle type was from traffic, not dust (line 360-362).

Line 341 and 346: The authors need to say more about how they are making the case that certain amines are both from vegetation and traffic. In the text, they refer to DPA being from both sources, and here in the conclusions they refer to DEA.

The authors also proposed that DEA-containing particles were also both from vegetation and traffic with two directions (northwest and southwest); please see the two paragraphs from line 242.

Section 4: The conclusions do not summarize the conclusions made throughout the paper and should be expanded.

Please see the revised text.

Again, we appreciate the reviewer for the comments which helped the authors to improve this manuscript.

References

Chen, Y., Wenger, J. C., Yang, F., Cao, J., Huang, R., Shi, G., Zhang, S., Tian, M., and Wang, H.: Source characterization of urban particles from meat smoking activities in Chongqing, China using single particle aerosol mass spectrometry, Environ. Pollut., 228, 92-101, 10.1016/j.envpol.2017.05.022, 2017.

Dall'Osto, M., Beddows, D., McGillicuddy, E. J., Esser-Gietl, J. K., Harrison, R. M., and Wenger, J. C.: On the simultaneous deployment of two single-particle mass spectrometers at an urban background and a roadside site during SAPUSS, Atmos. Chem. Phys., 16, 9693-9710, 2016.

Grigoratos, T., and Martini, G.: Brake wear particle emissions: a review, Environ Sci Pollut Res Int, 22, 2491-2504, 10.1007/s11356-014-3696-8, 2015.

Pankow, J. F.: Phase considerations in the gas/particle partitioning of organic amines in the atmosphere, Atmos Environ, 122, 448-453, 10.1016/j.atmosenv.2015.09.056, 2015.

Qin, X., Pratt, K. A., Shields, L. G., Toner, S. M., and Prather, K. A.: Seasonal comparisons of single-particle chemical mixing state in Riverside, CA, Atmos Environ, 59, 587-596, 10.1016/j.atmosenv.2012.05.032, 2012.

[Figure]

Rehbein, P. J., Jeong, C. H., McGuire, M. L., Yao, X., Corbin, J. C., and Evans, G. J.: Cloud and fog processing enhanced gas-to-particle partitioning of trimethylamine, Environ Sci Technol, 45, 4346-4352, 10.1021/es1042113, 2011.

Thorpe, A., and Harrison, R. M.: Sources and properties of non-exhaust particulate matter from road traffic: a review, Sci Total Environ, 400, 270-282, 10.1016/j.scitotenv.2008.06.007, 2008. Zhang, G., Bi, X., Chan, L. Y., Li, L., Wang, X., Feng, J., Sheng, G., Fu, J., Li, M., and Zhou, Z.: Enhanced trimethylamine-containing particles during fog events detected by single particle aerosol mass spectrometry in urban Guangzhou, China, Atmos Environ, 55, 121-126, 10.1016/j.atmosenv.2012.03.038, 2012.
* * *

---

## Editor Decision (ED1)

**Editor comments**

l. 36: '…that amines were undergone uptake by particles' should be replaced by 'that amines were taken up by particles'

l. 71/72: I think Referee #1 criticized this sentence because of its structure. I suggest removing it.

l. 72 – 82: This discussion of previous results is very brief. Add some more details in order to make it a useful comparison to your work:

> - Which particle types were found in Barcelona?

> - Why were TMA particles in Guangzhou important?

> - Did Zhang et al find TMA formation in fog? (It is not clear what 'similar scenario' refers to.)

l. 110: remove 'yielded'

l. 121-123: The sentence is still not clear. What are duplicate particles? 'Query results' is not clear either.

l. 134/135: This sentence is not clear: Do you mean: 'The RPA of DEA for each particle type was first calculated from all particles'? Please clarify.

l. 140: Do you mean '…accounting for 70% and 78% of all amine-containing particles in winter and summer…''?

l. 147: remove 'mixing'

l. 163/164: I think this sentence ('Seasonal variations …') can be removed here as the discussion of the unscaled size distribution starts only in l. 176.

l. 217: Only the left panel in Figure 3 shows wintertime results. If you refer to Figure 3a, you should add 'a)'and 'b)' to the Figure, respectively.

l. 252: Remove 'In addition'.

l. 271: It is not clear what you mean by 'this kind of discussion'.

l. 281-283: Remove this new sentence as it is already in l. 132. You can refer here simply to Section 2.3. (Even though I don't think it is necessary at all.)

l. 299-301: As Referee#2 stated correctly, you cannot make any assumption on aerosol water content and pH since you analyzed dry particles. Thus, the new sentence does not make much sense.

I suggest changing your text

*In this study, the relative acidity of amine-containing particles ((sulfate +nitrate)/ammonium, (Yao et al., 2011)) was in a range of 20-150, providing favorable conditions for the dissolution of DEA. Indeed, due to the nature of SPAMS, the amount of aerosol water content and pH were unavailable, making it difficult for further analysis.*

as follows (or similar):

*As particles are dried in the SPAMS, the amount of aerosol water content and pH were unavailable. The values of the anion/ cation ratio ((sulfate +nitrate)/ammonium, (Yao et al., 2011)) were in a range of 20-150 suggesting that the particles might have been acidic which favors the dissolution of DEA.*

l. 309: remove 'content'

l. 342: replace 'OC' by 'organic'

Supporting information:

- Use the updated manuscript title on the first page

- Change 'Table 1' to 'Table S1'

- Figure S1: This map is not very useful. The first panel does not show any contrast and, thus, does not give any information. I suggest adding a map that shows the location of the sampling site within China and/or the larger region. If you have a satellite image with high resolution, it might be useful to add to show the topology, together with a scale.

---

## Author Response (AR2)

Dear Prof. Barbara Ervens,

The authors are grateful for the Editor Comments. These comments are very helpful for us to improve our manuscript. We would thank everything you did in handling this manuscript. We will respond to the comments point-by-point.

Best wishes,

Yang Chen, on behalf of all authors

**Editor comments**

l. 36: '…that amines were undergone uptake by particles' should be replaced by 'that amines were taken up by particles'

Accepted and changed.

l. 71/72: I think Referee #1 criticized this sentence because of its structure. I suggest removing it.

Accepted and changed.

l. 72 – 82: This discussion of previous results is very brief. Add some more details in order to make it a useful comparison to your work: (Dall'Osto et al., 2013)

- Which particle types were found in Barcelona?

In Barcelona, five unique types of amine-containing particles were observed, including *amine-POA58* (composed of amines, sulfate, and nitrate), amine-*EST84*(environment tobacco smoke), *amine-SOA59* (composed of TMA and organics), *amine-SOA114*, and organic nitrogen amines (Dall'Osto et al., 2016; Dall'Osto et al., 2013).

- Why were TMA particles in Guangzhou important?

In Guangzhou, TMA-containing particles were important, taking up to 7% in number fraction during clear days and 35% during fog events (Zhang et al., 2012).

- Did Zhang et al find TMA formation in fog? (It is not clear what 'similar scenario' refers to.)

Zhang et al. found that, during fog events, the number fraction of TMA-containing particles took up to 35%; in size range of 0.5-2.0 μm, the fraction accounted up to 60% (Zhang et al., 2012).

We have improved this part, please see line 74-86.

l. 110: remove 'yielded'

Accepted and changed.

l. 121-123: The sentence is still not clear. What are duplicate particles? 'Query results' is not clear either.

Please see the revised version (Line 125-129):

The query strategy resulted in duplicate particles in the result when various amines co-existed in one single amine-containing particle. After the duplicate particles were removed from the multiple query results described above, all amine-containing particles were combined into an amine-containing particle cluster.

l. 134/135: This sentence is not clear: Do you mean: 'The RPA of DEA for each particle type was first calculated from all particles'? Please clarify.

There are two steps for calculating the RPA of DEA: the RPA of DEA were extracted in each particle, then the extracted RPAs were summed up.

Please see the revised version (Line 140-141):

To calculate the overall RPA of DEA, the relative peak areas of DEA in each particle were extracted and then summed up.

l. 140: Do you mean '…accounting for 70% and 78% of all amine-containing particles in winter and summer…''?

Yes, we have changed that. We would thank the Editor for the clarity (Line 146-148).

l. 147: remove 'mixing'

It has been changed.

l. 163/164: I think this sentence ('Seasonal variations …') can be removed here as the discussion of the unscaled size distribution starts only in l. 176.

The sentence has been removed.

l. 217: Only the left panel in Figure 3 shows wintertime results. If you refer to Figure 3a, you should add 'a)'and 'b)' to the Figure, respectively.

The reference of Figure 3a and b belonged to an old version; we have changed this to Figure 3 (Line 223).

l. 252: Remove 'In addition'.

It has been changed (Line 258).

l. 271: It is not clear what you mean by 'this kind of discussion'.

We have changed it to "the effect of RH on aerosol chemical processing."(Line 277-278)

l. 281-283: Remove this new sentence as it is already in l. 132. You can refer here simply to Section 2.3. (Even though I don't think it is necessary at all.)

We agree, and the sentence has been removed.

l. 299-301: As Referee#2 stated correctly, you cannot make any assumption on aerosol water content and pH since you analyzed dry particles. Thus, the new sentence does not make much sense.

*In this study, the relative acidity of amine-containing particles ((sulfate +nitrate)/ammonium, (Yao et al., 2011)) was in a range of 20-150, providing favorable conditions for the dissolution of DEA.*

*Indeed, due to the nature of SPAMS, the amount of aerosol water content and pH were unavailable, making it difficult for further analysis.*

*as follows (or similar):*

*As particles are dried in the SPAMS, the amount of aerosol water content and pH were unavailable.*

*The values of the anion/ cation ratio ((sulfate +nitrate)/ammonium, (Yao et al., 2011)) were in a range of 20-150 suggesting that the particles might have been acidic which favors the dissolution of*

*DEA.*

Yes, we have replaced the map in the SI.

l. 309: remove 'content'

We have changed (Line 313).

l. 342: replace 'OC' by 'organic'

We have changed (Line 346).

Supporting information:

- Use the updated manuscript title on the first page

Yes, we have changed this part.

- Change 'Table 1' to 'Table S1'

Yes, we have changed this part.

- Figure S1: This map is not very useful. The first panel does not show any contrast and, thus, does not give any information. I suggest adding a map that shows the location of the sampling site within China and the larger region. If you have a satellite image with high resolution, it might be useful to add to show the topology, together with a scale.

We have changed the map of the sampling site.

References

Dall'Osto, M., Querol, X., Alastuey, A., Minguillon, M. C., Alier, M., Amato, F., Brines, M., Cusack, M., Grimalt, J. O., Karanasiou, A., Moreno, T., Pandolfi, M., Pey, J., Reche, C., Ripoll, A., Tauler, R., Van Drooge, B. L., Viana, M., Harrison, R. M., Gietl, J., Beddows, D., Bloss, W., O'Dowd, C., Ceburnis, D., Martucci, G., Ng, N. L., Worsnop, D., Wenger, J., Mc Gillicuddy, E., Sodeau, J., Healy, R., Lucarelli, F., Nava, S., Jimenez, J. L., Gomez Moreno, F., Artinano, B., Prévôt, A. S. H., Pfaffenberger, L., Frey, S., Wilsenack, F., Casabona, D., Jiménez-Guerrero, P., Gross, D., and Cots, N.: Presenting SAPUSS: Solving Aerosol Problem by Using Synergistic Strategies in Barcelona, Spain, Atmos. Chem. Phys., 13, 8991-9019, 10.5194/acp-13-8991-2013, 2013.

Dall'Osto, M., Beddows, D. C. S., McGillicuddy, E. J., Esser-Gietl, J. K., Harrison, R. M., and Wenger, J. C.: On the simultaneous deployment of two single-particle mass spectrometers at an urban background and a roadside site during SAPUSS, Atmos Chem Phys, 16, 9693-9710, 10.5194/acp-16-9693-2016, 2016.

Zhang, G., Bi, X., Chan, L. Y., Li, L., Wang, X., Feng, J., Sheng, G., Fu, J., Li, M., and Zhou, Z.: Enhanced trimethylamine-containing particles during fog events detected by single particle aerosol mass spectrometry in urban Guangzhou, China, Atmos Environ, 55, 121-126, 10.1016/j.atmosenv.2012.03.038, 2012.